# Using stated-preferences methods to develop a summary metric to determine successful treatment of children with a surgical condition: a study protocol

Oliver Rivero-Arias ,[1] John Buckell ,[2] Benjamin Allin ,[1,3] Benjamin M Craig,[4] Goher Ayman ,[1] Marian Knight ,[1] the CSOR Collaborative Group

[1]National Perinatal Epidemiology Unit, Nuffield Department of Population Health, University of Oxford, Oxford, UK
[2]Health Economics Research Centre, Nuffield Department of Population Health, University of Oxford, Oxford, UK
[3]Chelsea and Westminster Hospital, London, UK
[4]Department of Economics, University of South Florida, Tampa, Florida, USA

**Correspondence to**
Dr Oliver Rivero-Arias;
oliver.rivero@npeu.ox.ac.uk

## ABSTRACT

**Introduction** Wide variation in the management of key paediatric surgical conditions in the UK has likely resulted in outcomes for some children being worse than they could be. Consequently, it is important to reduce unwarranted variation. However, major barriers to this are the inability to detect differences between observed and expected hospital outcomes based on the casemix of the children they have treated, and the inability to detect variation in significant outcomes between hospitals. A stated-preference study has been designed to estimate the value key stakeholders place on different elements of the outcomes for a child with a surgical condition. This study proposes to develop a summary metric to determine what represents successful treatment of children with surgical conditions.

**Methods and analysis** Preferences from parents, individuals treated for surgical conditions as infants/children, healthcare professionals and members of the public will be elicited using paired comparisons and kaizen tasks. A descriptive framework consisting of seven attributes representing types of operations, infections treated in hospital, quality of life and survival was identified. An experimental design has been completed using a D-efficient design with overlap in three attributes and excluding implausible combinations. All participants will be presented with an additional choice task including a palliative scenario that will be used as an anchor. The survey will be administered online. Primary analysis will estimate a mixed multinomial logit model. A traffic light system to determine what combination of attributes and levels represent successful treatment will be created.

**Ethics and dissemination** Ethics approval to conduct this study has been obtained from the Medical Sciences Inter-Divisional Research Ethics Committee (IDREC) at the University of Oxford (R59631/RE001-05). We will disseminate all of our results in peer-review publications and scientific presentations. Findings will be additionally disseminated through relevant charities and support groups and professional organisations.

## STRENGTHS AND LIMITATIONS OF THIS STUDY

⇒ Our descriptive system has been developed using formative research best practice guidance.
⇒ Our experimental design employs attribute level overlap and exclude implausible combinations of attribute and levels in the context of children with a surgical condition.
⇒ All participants will complete a choice task paired comparison with one of the alternatives representing a palliative scenario that will be used as an anchor.
⇒ Preference data from different stakeholders relevant to the decision context will be available to estimate the final summary metric.
⇒ Given that children with surgical conditions are relatively few in number, data collection may present challenges, in particular for the identification of parents and healthcare professionals, which will be mitigated using a thorough recruitment strategy.

provide specialised surgery in children.[1] Specialised surgery in children includes:

a. Management of rare surgical conditions in children.
b. Provision of specified specialised surgical procedures during childhood.
c. Surgery in neonates.
d. Surgical management or procedures for more common paediatric surgical conditions when a child requires specialist preoperative, anaesthetic or postoperative care (simple surgical procedures in children with complex medical needs).

Many of the conditions falling under the remit of specialised surgery in children commissioning affect only a few hundred children in England and Scotland each year, and what little is known about these children's long-term health and well-being suggests that even after treatment, they have significant ongoing healthcare needs.[2–8] Widespread variation in the management of children with

## INTRODUCTION

In England and Scotland, there are currently 24 Trusts/Health Boards commissioned to

these conditions currently occurs,[2–7] but due to the rarity of the conditions, it has not to date been possible to identify how much of this variation is unwarranted or associated with variation in outcome. To identify unwarranted variation in management and outcome between centres providing specialised surgery in children it is necessary to develop mechanisms that will:

1. Collect accurate, unbiased data about children treated in individual centres.
2. Combine data from children with different conditions in a way that enables meaningful outcomes analysis.
3. Enable adjustment for casemix factors affecting centres' outcomes.

The NIHR-funded study 'Improving unwarranted variation in outcomes of children's surgery through a new Children's Surgery Outcome Reporting system using routinely available data (CSOR)' investigates whether one unified system is capable of addressing these three issues and therefore reducing unwarranted variation in surgical care. This protocol paper describes the study design of one of the CSOR sub-studies, tackling the second barrier.

The gold-standard approach for comparing outcomes of interest to patients with a specific condition is to use a core outcome set (COS).[9] Several COSs have recently been developed that are relevant to children with a surgical condition.[10–13] Development of these COSs has highlighted that while there are attributes of successful treatment that are unique to each individual condition, there are also attributes that are common across conditions. A single outcome measure that combines these common attributes with attributes that accurately reflected the condition specific elements of each COS (eg, a measure of quality of life) would be able to provide a meaningful assessment of how successful the treatment of a child with any surgical condition has been. The aim of this study is therefore to develop an algorithm to assist in combining these attributes into one summary metric, and therefore, determining what constitutes successful treatment of children with a surgical condition. This metric would facilitate combination of data from children with different conditions enabling meaningful outcomes analysis.

Whether a certain combination of common core outcomes across conditions indicates a successful or unsuccessful treatment depends on the value that relevant stakeholders place on the different elements of the core outcomes. Economists employ preference elicitation techniques to determine such values.[14] Stated preference techniques such as discrete choice experiments (DCEs) are well suited to understand the value of potential combinations of core outcomes of paediatric surgery.[15]

A DCE is an experiment with choice tasks that elicit preferences indicating how individuals value attributes of alternatives in a decision scenario.[15] The value of a scenario depends on the levels associated to attributes, which are the characteristics of health, treatments or healthcare services being evaluated.[15] During a DCE, participants are presented with a number of scenarios and are asked to choose their preferred option, trading off among the attributes. DCEs can take different formats, but paired comparisons, where the participant is presented with two scenarios and asked to choose one, are most widely used.[16] Other alternatives to paired comparisons exist including best-worst scaling and more recently kaizen tasks.[17 18] Participants' choices in stated preferences exercises are analysed using discrete choice models, where choices are associated with combinations of attributes and levels to understand participants' preferences (ie, their relative importance).

This is a protocol for a stated-preference study designed to estimate the value key stakeholders place on different combinations of health and care outcomes following treatment of a child with a surgical condition. Stakeholders will be presented with a series of paired comparisons and novel kaizen tasks to elicit their preferences. The final product of the collected data will be an algorithm to determine whether the treatment of a child with a surgical condition has been 'successful' or 'unsuccessful'. The use of stakeholder preferences to help in healthcare decision making by policymakers has increased considerably across most developed jurisdictions.[16] This has been accompanied by best practice guidance for developing such studies, applied herein.[19–22]

## Aims

To understand how parents, guardians and healthcare professionals caring for children with a surgical condition, individuals treated for surgical conditions as children, and members of the general public value common health and care outcomes following treatment of a child with a surgical condition. Specifically:

- To estimate the relative importance of key health outcomes following treatment for a surgical condition in childhood for multiple conditions using a paired comparison and a novel kaizen task.

- To compare preferences between the two sources of preference data and type of participant in the context of children with a surgical condition.

- To estimate an algorithm using weights derived from the relative importance estimates to derive a summary metric that categorises outcomes following surgery in childhood into 'successful' or 'unsuccessful' outcome.

## METHODS AND ANALYSIS
### Overview of the framework for the stated-preference study

Figure 1 describes the framework and different phases that will be followed to conduct this study. This protocol describes the following sections: (1) identification and description, (2) experimental design, (3) survey instrument and (4) statistical evaluation.

### Identification and description
#### Decision model and descriptive framework

We have followed the most recent guidance on formative research for the identification of attributes and levels for

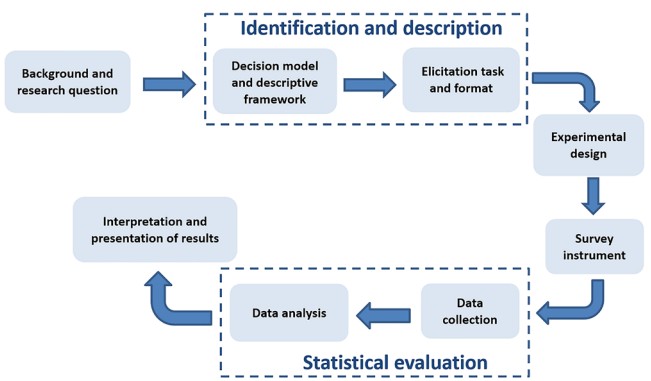

**Figure 1** A framework for stated-preference studies.

the descriptive system used in this study.[22] Our decision problem explores how to best conceptualise what constitutes a more successful outcome following treatment for a surgical condition in childhood from the values that relevant stakeholders place on key core outcomes across paediatric surgical conditions. Our decision model hypothesises that a successful treatment for a child with a surgical condition can be represented by a combination of characteristics or attributes. In this study, we define attributes as core health outcomes included in available COSs relevant to paediatric surgery. The attributes and associated levels that describe potential outcomes following treatment define our descriptive framework. We used literature reviews, interviews with parents and paediatric surgeons, and group discussions with our Parent Advisory Group (PAG) to determine the descriptive system for the final survey instrument. Our PAG consists of over 100 parents and family of children who have undergone early surgery for conditions including Hirschsprung's disease, gastroschisis, exomphalos, short bowel and necrotising enterocolitis.

An initial list of conceptual attributes were identified through a review of published COSs relevant to paediatric surgery. Relevant COSs have been developed for children with Hirschsprung's disease, gastroschisis and appendicitis, as well as for children receiving neonatal care in a high-income setting.[10–13] Each of these COSs was developed using a combination of literature reviews, an online Delphi process and consensus meetings, and included in their stakeholder groups, clinicians, allied health professionals, parents and children or adults previously treated for the target condition. We also conducted focused discussions with our PAG who had the opportunity to comment on the COSs identified in the literature review. The core outcomes identified in the above mentioned conditions are presented in table 1.

An iterative process to the identification of attributes was followed. In a first step, conducted by the clinical team (BA and MK), overlap in outcomes of importance was identified between the four COSs. Each COS also identified outcomes that were relevant only to the condition of interest, and not represented in the other COSs. In a second step, we reviewed these condition-specific

outcomes with our PAG in order to determine how best to represent them in the descriptive system. The group concluded that these condition-specific outcomes were highly likely to impact the child's overall quality of life and would therefore be adequately represented through an overarching attribute of quality of life. Three outcome categories, including survival, adverse events and quality of life, were common to all four relevant COSs and therefore selected as the initial set of attributes for the descriptive system. Similar outcomes have also been identified as important in other developed paediatric COSs.[23 24] The specific adverse events identified from each COS were also discussed with our PAG and the CSOR paediatric surgeons in a group discussion. The PAG and surgeons agreed that the main adverse events were better summarised as operations, and hospitalisations due to significant infections. Therefore, four core attributes were identified at the end of the second step. A description of each attribute is given next:

### Operations
Most surgical conditions are treated with one or more operations. Some operations that a child might undergo will be planned at the beginning of the child's treatment, while some will be emergency, or unplanned operations. The complexity of each operation also varies, from minor operations, such as draining an abscess, to more major operations, such as removing sections of intestine (bowel or gut). For the purpose of the preference study, the number of operations will be presented within four different attributes according to type of operation.

### Whether the child has an infection treated in hospital after their operation
Each of the COSs identified as relevant to paediatric surgery include condition-specific significant infective complications, such as enterocolitis and intra-abdominal abscess. Some also include a more generalised measure of significant infection, sepsis. For the purposes of the preference study, the infective complications included in each COS will be represented by the attribute infections treated in hospital, and the levels will define the frequency of infections.

### The child's quality of life
Each of the COSs included the outcome quality of life, while some also specifically included outcomes relating to psychological well-being. There are multiple instruments to measure quality of life in children, but they have not been validated in children with a surgical condition. These tools generally describe multiple domains, including social functioning, physical functioning and psychological well-being, with their output generally reported in a continuous manner. However, for the purposes of our descriptive system, quality of life will be categorised as good, fair or poor. The impact of key condition-specific outcomes will be reflected in the child's overall quality of life.

**Table 1** Summary of identified core outcomes in neonatal conditions

| Category | Core outcome | Included in core outcome set? | | | |
| --- | --- | --- | --- | --- | --- |
| | | Hirschsprung's disease | Gastroschisis | Neonatal care | Appendicitis |
| Survival | Survival | x | x | x | x |
| Quality of life | Quality of life | x | x | x | x |
| | Psychological stress | x | | | x |
| | Time away from full activity | | | | x |
| Adverse events | Unplanned reoperation | x | | | x |
| | No of operations | | x | | |
| | Severe gastrointestinal complication | | x | | |
| | Retinopathy of prematurity | | | x | |
| | Chronic lung disease | | | x | |
| | Bowel obstruction | | | | x |
| | Readmission | | | | x |
| | Length of hospital stay | | | | x |
| | Significant infection | | x | x | |
| | Hirschsprung's associated enterocolitis | x | | | |
| | Necrotising enterocolitis | | | x | |
| | Wound infection | | | | x |
| | Wound complication | | | | x |
| | Intra-abdominal abscess | | | | x |
| Condition specific | Faecal incontinence | x | | | |
| | Bowel function score | x | | | |
| | Voluntary bowel movements | x | | | |
| | Urinary incontinence | x | | | |
| | Permanent stoma | x | | | |
| | Growth | | x | | |
| | Time on parenteral nutrition | | x | | |
| | Liver disease | | x | | |
| | Brain injury on imaging | | | x | |
| | Motor/cognitive/visual/ hearing ability | | | x | |
| | Antibiotic failure | | | | x |
| | Negative appendicectomy | | | | x |
| | Recurrent appendicitis | | | | x |

*How long the child survives after their diagnosis*

Although death is relatively uncommon following most surgery in childhood, it is such a significant outcome that all four COSs relating directly to childhood surgery, and the majority of paediatric COSs include it. For the purposes of the descriptive system, this outcome will be presented positively as survival.

The identification of the attribute levels also employed an iterative process. First, we reviewed the epidemiological data available for each of the attributes to guide the range that could be presented to participants. Existing large-scale cohort studies describing the outcomes for children with a range of conditions treated under the specialised surgery in children commissioning, and affecting different regions of the body, modes of presentation and urgencies of intervention were reviewed.[2 4–6 8 25–33] A researcher (BA) extracted point estimates and associated measures of uncertainty for each of the attributes. In discussions with two other researchers (OR-A and BMC) initial deterministic ordinal levels for each attribute were developed. The selection of levels considered the potential participant cognitive burden and the ability to test appropriate functional forms hypothesis (eg, linear, quadratic) for quantitative attributes. This initial list of ordinal levels

**Table 2** Study descriptive framework

| Attributes | Attribute levels |
|---|---|
| Planned major operations related to the condition | ▶ No planned major operations<br>▶ One planned major operation<br>▶ Two planned major operations<br>▶ Six planned major operations |
| Planned minor operations related to the condition | ▶ No planned minor operations<br>▶ One planned minor operation<br>▶ Two planned minor operations<br>▶ Six planned minor operations |
| Emergency major operations related to the condition | ▶ No emergency major operations<br>▶ One emergency major operation<br>▶ Two emergency major operations<br>▶ Six emergency major operations |
| Emergency minor operations related to the condition | ▶ No emergency minor operations<br>▶ One emergency minor operation<br>▶ Two emergency minor operations<br>▶ Six emergency minor operations |
| Infections treated in hospital | ▶ No infections treated in hospital<br>▶ One infection treated in hospital<br>▶ Two infections treated in hospital<br>▶ Six infections treated in hospital |
| Child's quality of life | ▶ Good quality of life<br>▶ Fair quality of life<br>▶ Poor quality of life |
| How long the child survived after their diagnosis | ▶ More than 20 years, without any expectation that their surgical condition would shorten their life expectancy<br>▶ 20 years<br>▶ 5 years<br>▶ 1 year<br>▶ 6 months<br>▶ 1 month |

was discussed with paediatric surgeons in a group meeting to ensure their clinical appropriateness and also suggested changes to the wording. The language used to describe the levels was refined further following review by members of the PAG. Finally, we conducted three think aloud exercises with different parents who completed a mock choice task of the survey instrument with two of the researchers (OR-A and JB). Parents were given the opportunity during these interviews to comment on the wording used for the attributes and associated levels. The proposed attributes and attribute levels are described in the descriptive framework in table 2.

### Elicitation task and format

Two elicitation formats will be used in this study to estimate preferences: a paired comparison and a kaizen task. Each paired comparison will include two scenarios describing combinations of treatment outcomes of a child with a surgical condition without an opt-out option. An example is presented in online supplemental material 1. This type of choice task is the most widely used format in health preference research[16] and has been used previously to elicit preferences for outcomes of surgery.[34–36]

Recent work has reported that valuation can be problematic in the context of child health.[37 38] Eliciting values to inform decision making at the start of life or early childhood requires stakeholders to complete tasks from someone else's point of view. There is some evidence that when the tasks in an elicitation exercise refer to someone else's instead of their own preferences, individuals find the exercise strenuous. This is accentuated when the perspective is that of a newborn babies or a young individual.[37] In a paired comparison, it is easy to understand that choosing between two undesirable outcomes of paediatric surgery could be distressing to stakeholders. Moreover, members of the general public may find it both distressing and unfamiliar as they find it difficult to relate to the decision context.[39] This in turn can affect the preferences elicited in paired comparisons.

In addition to the paired comparisons in this study, we will therefore also administer a series of kaizen tasks to participants (online supplemental information 1). Each kaizen task begins with a paired comparison between a single profile and a palliative one (ie, no operations and no infections, but the child has fair quality of life and dies within 1 month). Next, the respondent makes three

improvements to the single profiles. After these improvements, the respondent completes a final paired comparison between the improved profile and a palliative one. Respondents may find the task of improving a child's health to be more engaging than choosing between two diverse outcomes.

### Experimental design

A preliminary experimental design has been completed for both elicitation exercises using a three-step approach: (1) generation of a design for the paired comparison task and (2) selection of pairs for the first and third part of the kaizen task and (3) selection of profiles for the second part of the kaizen task.

Our preliminary design has employed a D-efficient design to identify the combination of pairs to present in the paired comparison.[15] This is a procedure for generating choice tasks (for respondents) in a way that maximises the statistical efficiency of the choice models that will be estimated. Several restrictions were imposed including 'overlaps' in three attributes to reduce cognitive burden (similar attribute levels between pairs) and implausible combinations of outcomes. These combinations were identified by the CSOR surgical team and included:

a. Reject scenarios in which quality of life is good, the child survived for less than 1 year, and the child underwent 20 or more operations.
b. Reject scenarios in which the child underwent any number of minor operations, but no major operations, and survived for less than 20 years.
c. Reject scenarios in which the child underwent fewer than six minor operations, no major operations, and quality of life is low.
d. Reject scenarios in which quality of life is good, the child survived for less than 6 months, and the child underwent any major operations.

A candidate set was created including these restrictions and used as the initial candidates in Ngene.[40] We generated a preliminary design with 45 choice tasks divided into five blocks to which participants will be randomly allocated that is, nine choice tasks in each block. Participants will be randomised to one of these five blocks and the order of the pair in each choice task will also be randomised.

An additional choice task was added to all blocks, which serves as an 'anchor'. This anchor was a palliative profile defined as having no operations, no infections, fair quality of life and a survival of 1 month. This anchor will be used to facilitate comparisons between paired comparison and kaizen responses and also when developing the final algorithm for CSOR. The experimental design is presented in online supplemental material 2.

The experimental design for the kaizen task was constructed directly from the pairs in online supplemental material 2. Given each pair, an initial profile of the kaizen task was constructed from the worst attributes found in the pair. Likewise, the four possible improvements were

defined to be the best attributes found in the pair. Therefore, the preference path captured by the kaizen task should agree with the paired comparison response (ie, the path passes through profile chosen in the paired comparison before passing through the profile not chosen in the paired comparison).

### Survey instrument

The survey will be administered online and will be programmed in Oxford University servers with an open source platform. The survey will consist of an initial participant information and consent form, followed by a general welcome, three screening questions, an introduction to the research question and description of attributes (provided in both written and short video formats). For each attribute, respondents will answer warm-up tasks to give their view on the attribute for a hypothetical condition (see online supplemental material 1). This will be followed by the preference elicitations starting with the 10 paired comparisons and then the three kaizen tasks. For both tasks, participants will have the opportunity to complete a practice question. At the end of the elicitation tasks, participants will be asked three debriefing questions covering which exercise they found easier to complete, prefer to complete and easier to understand. Finally, a set of demographic questions will be collected including experience with neonatal/childhood surgical conditions, employment status and education qualifications. For healthcare professionals we will also ask their job title and level of professional experience with neonatal/childhood surgical conditions.

A preliminary mock survey has been completed and is presented in online supplemental material 1. In developing the instrument, preliminary testing was undertaken to maximise user understanding as described in Decision model and descriptive framework.

### Statistical evaluation

#### Data collection, recruitment strategy and sampling

The survey instrument will be completed by (1) a sample of parents of children with a surgical condition, (2) participants who had treatment for a surgical condition as a baby or a child, (3) health professionals caring for those who undergo surgery in childhood and (4) members of the general public. Main data collection will commence in October 2021.

Recruitment materials will be distributed via existing contacts for example, by our PAG, by the project's 'experts by experience', and by the project's healthcare professional team members; registers and mailing lists of support groups, charities and professional groups/bodies; and open advertising through support groups, charities and professional groups/bodies' communication channels for example, Twitter, Facebook, e-newsletters and websites. Distribution to health professionals will be via professional bodies. We will also include information on our project website and Twitter account, and advertise via Facebook to share the opportunity as widely as possible.

These recruitment strategies have been successfully used in previous quantitative and qualitative studies.[8 10–12 41]

We will use an online panel company to invite adult members of the general public to complete the survey. We will use quotas and a target recruitment strategy to ensure the sample is representative of the UK general population in terms of gender, age, social grade and nation.

In this study, we aim to collect 200 responses from parents and individuals with a surgical condition and 200 responses from healthcare professionals (total of 400) over a 3-month period. We are also aiming to collect 400 responses from members of the general public. This is based on typical sample sizes in the health literature and on simple minimum sample size principles.[42] Since there are no prior studies, it is not possible to obtain prior values for accurate power calculations. However, we will conduct a pilot study of around 80 individuals to assess whether our choice models will have sufficient power to detect significant differences.

The pilot study will also be used to assess the feasibility of the survey instrument in terms of finding programming errors and the process of data capture. Only participants recruited from the UK will be included in the final analysis.

## Data analysis

In the primary analyses, we will estimate a mixed multinomial logit model with individual-clustered standard errors. Unlike the conditional logit, the mixed specification relaxes the assumption of preference homogeneity by allowing individual-specific random parameters.[43] In the secondary analyses, we will explore preference heterogeneity further by estimating heteroscedastic and latent-class logit models. Although the secondary analysis of preference heterogeneity is more descriptive, we hypothesise that some respondents follow lexicographical processes and that respondents who complete the survey or task more quickly may exhibit greater variance in their responses than respondents who take more time to complete the survey or task. Other behaviours (eg, task sequence, object position) may also be related to stochastic processes.

In the primary analysis, we will test the following hypotheses:

1. Coefficients are logically consistent and with expected directions.
2. The coefficient for the anchor scenario is negative.
3. Survival and quality of life matter significantly more than the remaining attributes.
4. Survival matters more if child lives in good quality of life and survival matters less if child lives in poor quality of life.
5. Reductions in major planned or emergency operations matter more than reductions in minor or emergency operations.
6. We will observe differences among preferences from the different types of participants but not necessarily between elicitation tasks.

Each type of participant (parents, people who were treated for a surgical condition, healthcare professionals and members of the general public) will have completed the paired comparison and kaizen tasks. Therefore, eight sets of preference data will be available for analysis. We will compare preferences between types of respondent within each elicitation tasks using two approaches. First, latent scale coefficients of the attribute levels will be rescaled using the coefficient for the anchor scenario and predicted choice probabilities for types of respondent compared using mean square and absolute errors. Second, we will examine the relative attribute importance scores by attribute. This involves estimating the range of level combinations for each attribute and applying an attribute-based normalisation to enable comparisons.[44] A similar approach will be used to compare preference evidence between elicitation tasks.

The decision about which coefficients to use in the final algorithm for CSOR will be made by the CSOR Co-investigator Group based on the results of the primary and secondary analyses, feedback from participants and the face validity of the coefficients obtained.

For the final selected model with rescaled coefficients, we will predict the distribution of utilities of all possible combinations of attribute and levels. This distribution will be used to determine the likelihood of a combination to be considered successful or unsuccessful using a traffic light system: green area (high chance to be successful), amber area (uncertainty about success) and red area (not successful). The external validity of this algorithm will be evaluated in a separate study.

This study is expected to be completed by December 2022.

## Ethics and dissemination
### Ethical considerations
Ethics approval to conduct this study has been obtained from the Medical Sciences Inter-Divisional Research Ethics Committee (IDREC) at the University of Oxford (R59631/RE001-05). Informed consent will be obtained for all participants at the start of the survey.

### Dissemination
We will disseminate all of our results in peer-review publications and scientific presentations. A lay summary of the findings will be created using our PAG and circulated to parent support networks and the British Association of Paediatric Surgeons, via social media and on the project website.

## Patient and public involvement
Please see Identification and description.

**Acknowledgments** We are grateful to our Parental Advisory Group and the paediatric surgeons for their valuable contributions and help with the descriptive system. We would like to thank the participants of our think-aloud exercise, whose suggestions refined the final descriptive system and improved the online survey instrument. We are indebted to Lisa Hinton and Karolina Kuberska for their thoughtful comments on earlier versions of the survey instrument.

**Collaborators** Benjamin Allin (National Perinatal Epidemiology Unit, University of Oxford, and Chelsea and Westminster Hospital), Goher Ayman (National Perinatal Epidemiology Unit, University of Oxford), Timothy Bradnock (Royal Hospital for Children, Glasgow), John Buckell (Health Economics Research Centre, University of Oxford), Stefano Giuliani (Great Ormond Street Hospital), Nigel Hall (Southampton General Hospital and University of Southampton), Lisa Hinton (The Healthcare Improvement Studies (THIS) Institute, University of Cambridge), Crispin Jenkinson (Health Services Research Unit, University of Oxford), Simon Kenny (Alder Hey Children's Hospital NHS Foundation Trust), Marian Knight (National Perinatal Epidemiology Unit, University of Oxford), Karolina Kuberska (The Healthcare Improvement Studies (THIS) Institute, University of Cambridge), Kokila Lakhoo (Oxford University Hospitals NHS Foundation Trust, University of Oxford), Nick Lansdale (Royal Manchester Children's Hospital), Anna May Long (Cambridge University Hospitals NHS Foundation Trust), Alexander Macdonald (Chelsea & Westminster Hospital), Charles Opondo (National Perinatal Epidemiology Unit, University of Oxford), Rema Ramakrishnan (National Perinatal Epidemiology Unit, University of Oxford), Oliver Rivero-Arias (National Perinatal Epidemiology Unit, University of Oxford), Joanne Shepherd (Patient Representative), Gregor Walker (Royal Hospital for Children, Glasgow)

**Contributors** OR-A and MK developed the original idea for the study; OR-A, JB, BMC, BA and MK were responsible for the study design. OR-A, JB, BA and GA conducted the formative research component informing the development of attributes and levels for the descriptive system. JB, BMC and OR-A carried out the experimental design. OR-A and BMC implemented the online survey instrument. OR-A drafted the manuscript; all authors revised and approved the final manuscript. All members of the CSOR Collaborative Group participated in the overall programme design and the development and testing of the survey instrument.

**Funding** This study is funded by the National Institute of Health Research (NIHR) Health Services and Delivery Research (HS&DR) Programme (project reference NIHR 127481). MK is an NIHR Senior Investigator.

**Competing interests** None declared.

**Patient and public involvement** Patients and/or the public were involved in the design, or conduct, or reporting, or dissemination plans of this research. Refer to the Methods section for further details.

**Patient consent for publication** Not applicable.

**Ethics approval** Ethics approval to conduct this study has been obtained from the Medical Sciences Inter-Divisional Research Ethics Committee (IDREC) at the University of Oxford (R59631/RE001-05). Informed consent will be obtained for all participants at the start of the survey.

**Provenance and peer review** Not commissioned; externally peer reviewed.

**ORCID iDs**
Oliver Rivero-Arias http://orcid.org/0000-0003-2233-6544
John Buckell http://orcid.org/0000-0002-4157-4217
Benjamin Allin http://orcid.org/0000-0002-0575-2733
Goher Ayman http://orcid.org/0000-0002-5923-363X
Marian Knight http://orcid.org/0000-0002-1984-4575

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
