## [Reviewer comments · BMJ Open]

ARTICLE DETAILS

TITLE (PROVISIONAL)	Using stated-preferences methods to develop a summary metric to determine successful treatment of children with a surgical condition: a study protocol
AUTHORS	Rivero-Arias, Oliver; Buckell, John; Allin, Benjamin; Craig, Benjamin; Ayman, Goher; Knight, Marian; Collaborative Group, CSOR

VERSION 1 – REVIEW

REVIEWER	Oetzmann von Sochaczewski, Christina Universitätsklinikum Bonn, Sektion Kinderchirurgie der Klinik und Poliklinik für Allgemein-, Viszeral-, Thorax- und Gefäßchirurgie
REVIEW RETURNED	07-Dec-2021

GENERAL COMMENTS	Rivero-Arias et al. describe a study protocol for a stated-preference study to assess the value of different groups involved in the surgical care for children (and members of the public) and the relative value they attribute to aspects of the outcome. The study protocol includes all necessary items and adequately describes the relevant steps for the planned study in detail. However, some points require clarification, which I consider minor as they can be conducted by limiting the aims, narrowing the inclusion criteria, and an expansion of the discussion of the limitations. According to the title, the study protocol focuses on children with a surgical condition, whereas the abstract mentions "key paediatric surgical conditions", whereas the strengths and limitations again mentions any surgical condition, whereas the introduction then cites reports dealing with major congenital anomalies. The study design - dissemination via support groups etc. - also points towards the latter. As there are, to my knowledge, no patient support groups for patients that had an inguinal herniotomy as child and lived a happy life afterwards with a barely visible scar or for those that had a supracondylar humerus fracture treated with blount's method and now enjoy a full range of motion, it is unlikely that these (former) patients will be recruited in relevant numbers to the study. Moreover, patients with a simple and uncomplicated course will experience a considerable amount of cognitive dissonance as our colleagues from anaesthesiology frequently meet patients telling them they are fine and have no underlying medial conditions, although one can clearly see the scar from the sterniotomy. Consequently, the implied focus on major congenital anomalies (which also manifests itself in the provided items of the survey that only mention, apart from necrotising enterocolitis,
---

	major congenital anomalies) should be emphasised and the slightly overambitious aim to develop a core outcome set for all operations in children should be narrowed down to these chronic conditions. That said, the term key paediatric surgical conditions should be avoided, too, particularly if the authors would aim to include results from other high-income Western countries, which have a different spectrum of paediatric surgery: In Germany, we consider fractures of the growing bone a key aspect of paediatric surgery that cannot be left for orthopaedic surgeons, which is totally different in the United Kingdom. I am therefore also skeptical about including results from different healthcare systems in other high-income Western countries as they might just not be comparable as the differences are too large (exemplified in the literature for example in Volume 27 Issue 5 of the European Journal of Pediatric Surgery). These issues need to be resolved and this should preferably be done by a limitation towards major congenital anomalies (and can be achieved in the already started study by excluding those few respondents not dealing with major congenital anomalies). The recruitment via support groups etc. also raise another issue of selection bias apart from the implicit focus on major congenital anomalies: It is well-known that participation in online surveys particularly attract those with an ongoing interest in this matter, either because they are committed to it via the support groups or because they experienced a negative course of disease in the past. It would have been easily possible to partially quantify the extent of this bias by asking the respondents for the overall course of disease of the major congenital anomaly (e.g. rated on an ordinal scale). Therefore, I miss a discussion of the limitations arising from the recruitment of potential participants.
--	--

REVIEWER	Seyi-Olajide, Justina Lagos University Teaching Hospital, Surgery
REVIEW RETURNED	17-Dec-2021

GENERAL COMMENTS	This protocol and what it seeks to address sounds interesting however as the authors indicate, surgical conditions in children are so widely varied ranging from congenital anomalies to acquired conditions. It will be interesting to see to what extent the proposed protocol will address this. In addition, it is unclear how relevant this will be to real life paediatric surgical practice. I don't think that this protocol merits publication at this stage. The authors may want to publish the findings from their deployment of this protocol at which time adequate information may be available to make definite conclusions and recommendations
--

VERSION 1 – AUTHOR RESPONSE

Reviewer 1 (reviewers comments copied verbatim)

Comment 1: Rivero-Arias et al. describe a study protocol for a stated-preference study to assess the value of different groups involved in the surgical care for children (and members of the public) and the relative value they attribute to aspects of the outcome.

The study protocol includes all necessary items and adequately describes the relevant steps for the planned study in detail.

However, some points require clarification, which I consider minor as they can be conducted by limiting the aims, narrowing the inclusion criteria, and an expansion of the discussion of the limitations.

Response: we thank this reviewer for taking the opportunity to read our manuscript and we were pleased to read that all the comments were considered minor. We have made changes to our manuscript in light of the comments received, which we describe below.

Comment 2: According to the title, the study protocol focuses on children with a surgical condition, whereas the abstract mentions "key paediatric surgical conditions", whereas the strengths and limitations again mentions any surgical condition, whereas the introduction then cites reports dealing with major congenital anomalies. The study design - dissemination via support groups etc. - also points towards the latter. As there are, to my knowledge, no patient support groups for patients that had an inguinal herniotomy as child and lived a happy life afterwards with a barely visible scar or for those that had a supracondylar humerus fracture treated with blount's method and now enjoy a full range of motion, it is unlikely that these (former) patients will be recruited in relevant numbers to the study. Moreover, patients with a simple and uncomplicated course will experience a considerable amount of cognitive dissonance as our colleagues from anaesthesiology frequently meet patients telling them they are fine and have no underlying medial conditions, although one can clearly see the scar from the sternotomy. [...] These issues need to be resolved and this should preferably be done by a limitation towards major congenital anomalies (and can be achieved in the already started study by excluding those few respondents not dealing with major congenital anomalies).

Response: Conditions that are commissioned for treatment as part of the 'specialised surgery in children' pathway are either rare, or complicated/complex forms of more common conditions. This rarity and complexity means that in order to achieve the statistical power that is required for hospitals providing such services to understand whether their outcomes are as good as expected or not, it is necessary to combine outcomes for children across a broad range of surgical conditions. Some of these conditions will be major congenital anomalies, whilst some will be more common conditions that have had a complicated course. In order to achieve combination of outcomes across these conditions, it is necessary to first develop a single outcome measure that is clinically relevant and broadly applicable to a wide range of surgical conditions affecting children. The intention of this discrete choice experiment is that it will provide us with the information that we need to combine a series of seven attributes of treatment success which have previously been shown to be relevant to a wide range of paediatric surgical, paediatric and neonatal conditions, into one single outcome measure describing how successful the treatment of a child with a surgical condition has been.

Based upon this desired utility, the stated aim of this study is to "estimate an algorithm to derive a summary metric that categorises outcomes following surgery in childhood into "successful" or "unsuccessful" outcome." It is therefore intentional, and necessary, that recruitment to the study is broad. We will achieve such broad recruitment through a combination of Facebook and social media advertising to target those with experience of more common childhood conditions (appendicitis, hernia, etc.), and then recruitment through support groups and charities to ensure that rarer more complex conditions are also represented. In order to ensure clarity of the study rationale, we have amended the introduction. We have also sought to ensure that our target population is consistently described as 'children with a surgical condition' throughout the manuscript.

Comment 3: Consequently, the implied focus on major congenital anomalies (which also manifests itself in the provided items of the survey that only mention, apart from necrotising enterocolitis, major congenital anomalies) should be emphasised and the slightly overambitious aim to develop a core outcome set for all operations in children should be narrowed down to these chronic conditions.

Response: We would like to take the opportunity and clarify that this study is not attempting to develop a core outcome set. As described in the introduction, aims and methods, we are conducting a stated preference exercise using a discrete choice experiment. As per section 1.1, the aim of this stated preference exercise is:

To understand how parents, guardians and health care professionals caring for children with a surgical condition, individuals treated for surgical conditions as children, and members of the general public value common health and care outcomes following treatment of a child with a surgical condition. Specifically:

A. To estimate the relative importance of key health outcomes following treatment for a surgical condition in childhood for multiple conditions using a paired comparison and a novel kaizen task.

B. To compare preferences between the two sources of preference data and type of participant in the context of children with a surgical condition.

C. To estimate an algorithm using weights derived from the relative importance estimates in A to derive a summary metric that categorises outcomes following surgery in childhood into “successful” or “unsuccessful” outcome.

In order to avoid further confusion, we have extensively modified the introduction as described in response to comment 2 above.

Comment 4: That said, the term key paediatric surgical conditions should be avoided, too, particularly if the authors would aim to include results from other high-income Western countries, which have a different spectrum of paediatric surgery: In Germany, we consider fractures of the growing bone a key aspect of paediatric surgery that cannot be left for orthopaedic surgeons, which is totally different in the United Kingdom. I am therefore also skeptical about including results from different healthcare systems in other high-income Western countries as they might just not be comparable as the differences are too large (exemplified in the literature for example in Volume 27 Issue 5 of the European Journal of Pediatric Surgery).

Response: We have modified the introduction as per our response to comment 2 above to clarify the target population for this study. We hope this can address the reviewer’s concerns regarding different understandings of what constitutes paediatric surgery. Information will also be collected concerning the country of residence of study participants, and this will allow us to explore whether there are differences in preference according to territory. However, we confirm that preferences from different settings will not be combined and that the final algorithm will be developed using data from the UK.

Comment 5: The recruitment via support groups etc. also raise another issue of selection bias apart from the implicit focus on major congenital anomalies: It is well-known that participation in online surveys particularly attract those with an ongoing interest in this matter, either because they are committed to it via the support groups or because they experienced a negative course of disease in the past. It would have been easily possible to partially quantify the extent of this bias by asking the respondents for the overall course of disease of the major congenital anomaly (e.g. rated on an ordinal scale). Therefore, I miss a discussion of the limitations arising from the recruitment of potential participants.

Response: It may be possible that preferences from participants who do not complete our survey will be different from those who will complete it. The extent of this bias depends on how representative our participants will be of their respective groups. We will be collecting baseline demographic

information that will be compared with published data in the case of parents, people treated as child and health care professionals and with population estimates in the case of the general population. If we observe a potential selection bias, we will acknowledge this as a limitation when reporting the main results of the study. The reviewer has also suggested the addition of a visual analogue scale indicating the course of the disease associated to children and people treated as children as a measure to help identifying the extent of a potential bias. We discussed at the start of the project this issue and decided to include a question at the end of the survey about the quality of life and age of the child/people treated at the time of completion of the survey (see supplementary file 1). We think that such metrics may be more informative than the disease status as our preliminary analysis suggest that quality of life and survival are highly predictive of treatment success. We will report this information in a descriptive manner in our final manuscript to understand our sample representativeness.

Reviewer 2 (reviewers comments copied verbatim)

Comment 6: This protocol and what it seeks to address sounds interesting however as the authors indicate, surgical conditions in children are so widely varied ranging from congenital anomalies to acquired conditions. It will be interesting to see to what extent the proposed protocol will address this. Response: this comment was also made by reviewer 1 and we refer the editor/reviewer to our above comment 2 response.

Comment 7: In addition, it is unclear how relevant this will be to real life paediatric surgical practice. I don't think that this protocol merits publication at this stage. The authors may want to publish the findings from their deployment of this protocol at which time adequate information may be available to make definite conclusions and recommendations

Response: As per response to comment 2 from reviewer 1, we have modified the introduction to clearly describe the necessity for this study, and its application to real life paediatric surgical practice. From a wide range of previously published studies, it is clear that outcomes for children with surgical conditions are not as good as people commonly believe, or as good as they should be. It is only through fundamentally shifting the way that clinicians are able to interrogate their own outcomes, and the way in which research is conducted that steps will be made to improve outcomes. To our knowledge, this is the first time that anyone has attempted to develop a composite outcome measure based upon attributes identified from core outcome sets, and weights derived from a stated preference exercise, and therefore we believe that there is significant merit in allowing the wider community to read and critique the methodology we are using in a much more detailed format than would be possible in a manuscript publishing the results of the study. As is well recognised, it is also good practice to publish study protocols, particularly when novel methods are being used, in order to allow a thorough assessment of whether the published results of a study are likely to have been influenced by bias.

VERSION 2 – REVIEW

REVIEWER	Oetzmann von Sochaczewski, Christina Universitätsklinikum Bonn, Sektion Kinderchirurgie der Klinik und Poliklinik für Allgemein-, Viszeral-, Thorax- und Gefäßchirurgie
REVIEW RETURNED	08-Apr-2022
GENERAL COMMENTS	I don't have any additional comments.